# A Wide-Range-Response Piezoresistive–Capacitive Dual-Sensing Breathable Sensor with Spherical-Shell Network of MWCNTs for Motion Detection and Language Assistance

**DOI:** 10.3390/nano13050843

**Published:** 2023-02-24

**Authors:** Shuming Zhang, Xidi Sun, Xin Guo, Jing Zhang, Hao Li, Luyao Chen, Jing Wu, Yi Shi, Lijia Pan

**Affiliations:** Collaborative Innovation Center of Advanced Microstructures, School of Electronic Science and Engineering, Nanjing University, Nanjing 210093, China

**Keywords:** porous conducting polymer, strain sensing, dual sensing mode, breathable

## Abstract

It is still a challenge for flexible electronic materials to realize integrated strain sensors with a large linear working range, high sensitivity, good response durability, good skin affinity and good air permeability. In this paper, we present a simple and scalable porous piezoresistive/capacitive dual-mode sensor with a porous structure in polydimethylsiloxane (PDMS) and with multi-walled carbon nanotubes (MWCNTs) embedded on its internal surface to form a three-dimensional spherical-shell-structured conductive network. Thanks to the unique spherical-shell conductive network of MWCNTs and the uniform elastic deformation of the cross-linked PDMS porous structure under compression, our sensor offers a dual piezoresistive/capacitive strain-sensing capability, a wide pressure response range (1–520 kPa), a very large linear response region (95%), excellent response stability and durability (98% of initial performance after 1000 compression cycles). Multi-walled carbon nanotubes were coated on the surface of refined sugar particles by continuous agitation. Ultrasonic PDMS solidified with crystals was attached to the multi-walled carbon nanotubes. After the crystals were dissolved, the multi-walled carbon nanotubes were attached to the porous surface of the PDMS, forming a three-dimensional spherical-shell-structure network. The porosity of the porous PDMS was 53.9%. The large linear induction range was mainly related to the good conductive network of the MWCNTs in the porous structure of the crosslinked PDMS and the elasticity of the material, which ensured the uniform deformation of the porous structure under compression. The porous conductive polymer flexible sensor prepared by us can be assembled into a wearable sensor with good human motion detection ability. For example, human movement can be detected by responding to stress in the joints of the fingers, elbows, knees, plantar, etc., during movement. Finally, our sensors can also be used for simple gesture and sign language recognition, as well as speech recognition by monitoring facial muscle activity. This can play a role in improving communication and the transfer of information between people, especially in facilitating the lives of people with disabilities.

## 1. Introduction

With the progress of science and technology, various electronic products have been invented to provide help for human life and work. In this process, portable computers, smart watches, smart glasses and other wearable devices have been created and developed rapidly [1,2,3,4,5]. In the research of wearable devices, we focused on the skin, one of the largest and most important organs of the human body. Human skin is not only the first barrier against the outside world but also has many tactile receptors for sensing pressure, temperature, strain and other external stimuli [6,7]. By mimicking the sensory capabilities and characteristics of natural skin, a great deal of research has been conducted to develop bio-inspired electronic skin [3,8,9,10,11], which has important applications in wearable healthcare devices [12,13], smart robots [14,15], implantable medical devices [16], motion monitoring [17,18] and environmental perception [19,20]. As an important part of electronic skin, stress sensors based on different principles have been widely studied, including transistor sensing [21], capacitive sensing [22], piezoresistive sensing [23], piezoelectric sensing [24] and triboelectric sensing [25]. Piezoresistive and capacitive strain sensors are widely studied by researchers because of their simple preparation process, low cost and excellent flexibility [18,23,26,27,28,29,30,31,32]. On the other hand, compared with metal-based and semiconductor-based stress sensors, conductive polymer composite stress sensors based on percolation thresholds are one of the best choices for the next generation of electronic skin stress sensors due to their superior strain-sensing capability and their unique flexibility and light weight [8,33]. Breathability sensors provide comfortable contact with the human body by balancing the temperature and humidity of the human skin through the exchange of gases between the skin and the outside world [34]. However, conventional polymeric strain sensors have a single sensing mode, and it is difficult to provide a wide range of detection, high sensitivity and excellent response stability while maintaining soft and breathable properties using these devices. This greatly limits their application in various situations. In order to solve these problems, researchers have designed a variety of different microstructures in recent years, such as pyramid array structures [22,35], fiber structures [4,30,31,36], microcrack structures [33,37,38], sphere structures [39] and porous structures [26,32,40,41]. Polymer foam with a porous structure has been widely used in the sensor field due to its low density, large specific surface area, good compression recovery performance, simple preparation process and other advantages [23]. Meanwhile, due to their high electrical conductivity, large surface area and chemical stability, MWCNTs have been investigated in a variety of frontier sensing applications, such as optoelectronic sensors [42,43], medical sensors [44,45], chemical sensors [46,47], mechanical sensors [48] and nano semiconductor devices [49]. Among them, conducting polymer sensors doped with MWCNTs have received much attention in the field of mechanical sensing due to their fast response and high sensitivity. In addition, devices with multiple sensing mechanisms are important for artificial limbs and robots that need to be able to sense fine motion and environmental information. A variety of single-signal haptic sensors have been developed, piezoresistive and capacitive-based sensors can independently enable the sensing of pressure or temperature information. There are also e-skin systems that integrate multiple sensors to further improve the perception of the environment and motion [20,50]. In addition, special functional properties such as self-healing, self-powering, biodegradability, biocompatibility and breathability have been gradually integrated into these devices to obtain e-skins with comprehensive performance for practical applications [51,52,53]. In particular, breathability is an important way to improve the comfort of wearable devices by balancing the thermal humidity between the human body and the external environment. Therefore, haptic sensors with breathability are in high demand and of great relevance in wearable health monitoring, biomedical monitoring and implantable device applications [54].

In this paper, a novel strain sensor with a porous structure and a three-dimensional spherical shell network composed of MWCNTs is proposed, in which a dissolvable crystal was used as a sacrificial template. Firstly, the embedded three-dimensional spherical-shell network of MWCNTs enabled the sensor to have both piezoresistive and pressure-capacity-sensing capabilities, resulting in more accurate sensing data. In addition, the porous structure prepared by the dissolvable crystal as a sacrificial template gave the sensor a high sensitivity of −1.2/kPa (piezoresistive) and 0.38/kPa (pressure capacitive). With a wide pressure sensing range (1–520 kPa), it fully met the requirements of human health monitoring applications. In addition, the porous structure gave the sensor good comfort and skin affinity for adapting to different human application scenarios. Finally, a porous-structure sensor based on doped MWCNTs was applied to the human body to monitor physiological signals and joint movements, including muscle movement, finger flexion, elbow movement, knee movement and plantar pressure.

## 2. Materials and Methods

### 2.1. Materials

Commercia PDMS (Sylgard 184, Dow Corning Co., Midland, MI, USA) and a curing agent (Dow Corning Co., USA) were purchased from Dow Materials Sciences and were used as the polymer matrix in this study. Hydroxylated multi-walled carbon nanotubes (10–20 µm) were purchased from Jiangsu Xianfeng Nano Company (Nanjing, China) and were used as conductive fillers. Refined sucrose crystals were purchased from Taikoo to construct the cavity structure.

### 2.2. Fabrication of Sphere-Shell Three-Dimensional Structure of MWCNTs

A three-dimensional spherical-shell network of MWCNTs was prepared using refined sucrose crystal as a removable skeleton. MWCNTs with a mass ratio of 1% were first mixed with refined sucrose particles, which were stirred in a blender for 30 min to make the MWCNTs evenly coated on the surface of the sucrose crystals. Finally, an MWCNT network with a three-dimensional spherical-shell structure was formed by allowing the sample to stand at room temperature for 60 min.

### 2.3. Fabrication of CNT–PDMS Sponges

Highly compressible CNT–PDMS sponges were prepared by a simple sacrificial template method. First, 20 g of Dow Corning PDMS (Sylgard 184, Dow Corning Co. USA) was mixed with a curing agent in a ratio of 10:1 and mixed with carbon nanotubes (200 mg, Xianfeng, China). After removing the bubbles captured during agitation under mild vacuum conditions, the refined sucrose particles, which were evenly wrapped in the three-dimensional spherical-shell structure of the MWCNTs, were placed into a PDMS mixture that could penetrate the porous structure through a vacuum suction process and capillary forces. The pore size was determined by the grain size of the sugar. Then, the sample was cured at 100 °C under atmospheric pressure for 1 h. After curing, the sugar was dissolved in an ultrasonic bath in hot water for 1 h, and finally the sample was dried overnight. Then, two conductive copper tapes were assembled on the upper and lower surfaces of the prepared porous PDMS sponge with MWCNTs as electrodes, and the capacitance and resistance responses under corresponding compression were recorded.

### 2.4. Characterization

The surface morphology of the sensor was observed using an on-site scanning electron microscope (Nova Nano SEM 230, FEI. Hillsboro, Oregon, USA) at an operating voltage of 5 kV. All specimen thicknesses were confirmed by measuring the cross-section of the specimens. Considering the PDMS insulator, the samples were sputtered with a thin Au layer on the cross-section of the samples before SEM observation. The internal shape of the device was observed by an Ultra-Depth Three-Dimensional Microscope (KEYENCE VHX-6000. Osaka, Japan) with a 40× magnification of the cross-section. A series of strain and pressure tests were carried out on the microcomputer, a strain cycle was applied to the sensor by a ZHIQU ZQ-990 (ZHIQU. Dongguan, Guangdong, China) pressure stress tester and the capacitance and resistance signals of the sensor were detected by a KEYSIGHT E4980AL LCR (KEYSIGHT. Santa Rosa, CA, USA) tester at an operating voltage of 20 V and an operating frequency of 20 kHz. The information was collected and recorded by the software developed by LABVIEW.

## 3. Results

### 3.1. Structural Design and Sensing Principle of Dual-Mechanism Pressure Sensor

As shown in Figure 1a, MWCNTs and polydimethylsiloxane (PDMS) were used as pressure-sensitive materials to fabricate the pressure sensor. After the structural optimization processing and design integration of the MWCNTs, a wearable flexible pressure sensor with excellent breathability and flexibility was realized. The pressure sensor can be directly attached to human skin for the continuous monitoring of various stress–strain signals without causing any damage to the skin, and its schematic diagram is shown in Figure 1b. The entire pressure sensor does not require an encapsulation layer and substrate typical of conventional pressure sensors.

The porous structure of the pressure-sensitive layer was made of fine sugar wrapped in MWCNTs, and finally the fine sugars contained in the pressure-sensitive layer were treated with deionized water. The preparation methods and processes detail of the specific materials and devices are shown in Appendix A. Figure 1c shows the surface SEM images of the porous structure of the pressure-sensitive layer. The surface morphology of the sensor was observed using an on-site scanning electron microscope (Nova Nano SEM 230, FEI) at an operating voltage of 5 kV. The thicknesses of all the samples were determined by measuring the cross-sections of the samples. Due to the insulating properties of PDMS, a thin gold layer was sputtered onto cross-sections of the samples prior to the SEM observation. As can be seen in Figure 1c, the pores formed by the fine sugar were uniformly distributed throughout the PDMS layer. The internal morphology was observed using an Ultra-Depth Three-Dimensional Microscope (KEYENCE VHX-6000) at a 40× magnification, as shown in Figure 1d. The presence of the pores greatly increased the specific surface area of the entire pressure-sensitive layer, increased the contact area of the MWCNTs in the pressure-sensitive layer and provided the change in the conductive path inside the piezoresistive sensor, which is an effective way to achieve a high sensitivity over a wide sensing range.

### 3.2. Breathability and Comfort

The breathability and flexibility of wearable pressure sensors have remarkable effects in health monitoring. As shown in Figure 2a, the MWCNT-based pressure sensors could be directly attached to human skin. This was due to the overall porous structure of the pressure-sensitive layer, and when attached to human skin, the epidermal temperature and sweat from the human body could be quickly dissipated into the surrounding environment through the pressure sensor. In addition, as seen in Figure 1c, the sensor had a large number of cracks and porous structures, which have the advantage of optimizing their surface coverage while maintaining breathability. The properties of this structure could provide great advantages in terms of the design and construction of medical devices for physiological signal monitoring.

To demonstrate the permeability of the sensor, the permeability performance of the pressure sensor was evaluated using the heat dissipation rate of the human epidermis, and the optical photographs of the process are shown in Figure 2b. The skin epidermis temperature of bare leaky skin, the skin temperature of the porous pressure sensor covered with MWCNTs, the skin epidermis temperature covered with a PDMS film and the epidermis temperature covered with band-aids were tested separately. After 360 min, the epidermal temperature of the skin covered with the MWCNT-based porous pressure sensor was almost the same as that of the bare epidermal skin temperature. In contrast, the skin surface covered by PDMS showed a significant increase in temperature and was accompanied by redness of the skin surface. The skin epidermal temperature was recorded by IR thermometer (Appendix A) and is shown in Figure 2c, and comparative photographs of the skin are shown in Figure 2b. In addition, we used the rate of water vapor molecules penetrating the film to evaluate the permeability of the porous pressure sensor, and an optical photograph of the process is shown in Figure 2d. Glass bottles holding colored water were sealed by various functional films including para film, PDMS film, PET film, paper and porous MWCNT film. Afterwards, they were compared to open glass bottles that also held the same mass of water at room temperature. After 15 days, the level of water evaporation in the glass bottle sealed with the porous MWCNT film was second only to that of the unsealed control group. In contrast, the mass of water in the glass bottles encapsulated by the other films remained almost constant. The weight loss of water was recorded every 24 h throughout the experiment (Figure 2e), and photographs of the water weight loss process are shown in Appendix A. Based on these results, the device exhibited excellent permeability.

### 3.3. Electrical Output Performance

Stress sensors for monitoring physiological signals and movements of human joints require their collected data to have a high accuracy and stability. In order to improve the accuracy of the data, the sensitivity and linear sensing range of the pressure sensor were tested using the dual sensing mode. The test instrument shown in Figure 3a was used to test the change in the resistance and capacitance of the composite porous-structure strain sensor under different strains at an operating voltage of 20 V and an operating frequency of 20 kHz. In order to optimize the range and sensitivity of the sensor, different MWCNT-doping concentrations were tested (Appendix A). It was found that when the mass ratio of the MWCNTs inside the PDMS skeleton was 1% and the mass ratio of the MWCNTs attached to the surface of the porous structure was 1%, the piezoresistive performance of the sensor provided excellent uniformity under pressure application and release while providing an optimal detection range and sensitivity (Appendix A). The capacitance and resistance values of the sensor under a load pressure ranging from 1 kPa to 520kPa, are shown in Figure 3b. The pressure-sensitive layer of the sensor had opposite resistance and pressure changes throughout a pressure cycle (1–520 kPa), while its capacitance and pressure changed with the same trend. As shown in Figure 3c, the sensor demonstrated a good consistency in terms of boosting and lowering the pressure within the range of 1~520 kPa, and it had good elasticity and stability. We introduced the sensitivity (S) and gauge factor (GF) in order to characterize and discuss the sensitivity of the sensor. In detail, S is defined as S = δ(ΔR/R0)/δP, where ΔR/R0 is the relative capacitance change, and P is the applied pressure, which corresponds to the slope of the pressure–response curve in Figure 3c. At the first stage (1–20 kPa), the porous structure began to deform, the number of conductive paths increased rapidly and the distance of the conductive paths shortened. At this time, the maximum sensitivity was −1.2%/kPa. At the second stage (20 kPa–130 kPa), the deformation rate of the porous structure decreased, and the sensitivity was −0.42%/kPa. At the third stage (130 kPa–520 kPa), the sensitivity was −0.053%/kPa, and the porous structure no longer generated deformation. At this time, the deformation of the sensor was mainly caused by the deformation of the PDMS itself, the number of conductive paths was basically unchanged and the conductive paths continued to shorten. For the capacitive response, the sensor capacitance increased with the increase in the sensor strain degree. When pressure was applied to the surface of the device, the device underwent vertical deformation, and the spherical holes inside the device gradually closed, increasing the conductive path and reducing the resistance. The capacitance of the device was provided by both the PDMS skeleton structure and the air inside the spherical cavity, while the PDMS doped with MWCNTs inside it had a higher dielectric constant than air. When the sensor was compressed, the cavity was compressed, the air inside the cavity was reduced and the PDMS ratio was increased, resulting in an increase in the sensor’s capacitance [32]. When the applied pressure was removed, the vertical deformation of the device disappeared, and the spherical cavity inside the device gradually recovered, leading to a reduction in the conductive path and an increase in the resistance. At this point, the proportion of air increases, resulting in a decrease in the capacitance. Due to the capacitance structure formed by the microcavities inside the sensor and the MWCNT layer on its surface, the critical points at the different stages of capacitance change were almost the same as the critical points at the different stages of the strain of the sensor, resulting in a three-stage process similar to the resistance response. As shown in Figure 3d, we explored the sensitivity of the sensor to strain. ΔR/R0 was estimated by the formula ΔR/R0 = −3.43ε + 0.03ε^2^ − 0.08(%) (Appendix A), and ΔC/C0 was estimated by the formula ΔC/C0 = 1.97ε − 0.009ε^2^ + 0.08(%) (Appendix A), where ΔR/R0 is the change in the relative resistance, ΔC/C0 is the change in the relative capacitance and ε is the strain applied, as shown in Figure 3d. Therefore, the corresponding gauge factor (GF) showed a linear trend (GF (resistance)= 0.06ε − 3.43 (%), GF (capacitance) = 1.97 − 0.018ε (%)). It turned out that the strain sensor was sensitive and comparable to some other piezoresistive sensors (Appendix A). The strain of the sensor under pressure is shown in Figure 3e. At 1–130 kPa, the strain was mainly generated by the porous structure, and at 130kPa–520kPa, the strain was mainly generated by the PDMS matrix.

We compared the pressure-sensing performance of the prepared porous MWCNT-based pressure sensors with that of some other devices mentioned in the literature, and Appendix A lists more specific comparisons of the performance between these pressure sensors mentioned in the literature. Some pressure sensors have a large stress monitoring range but a low sensitivity. Some sensors have a high sensitivity but a narrow and nonlinear detection range. Moreover, most conventional sensors are single-physical-quantity sensing, whereas the dual-sensing-mode pressure sensor produced in this work had a relatively high sensitivity in the high-stress detection range, which is a great advantage compared to conventional wearable electronic devices. As shown in Figure 3c, the highest sensitivity of this pressure sensor was 0.38/kPa (capacitance) and −1.2/kPa (resistance) over a wide detection range of 1–520 kPa. It is noteworthy that the MWCNT-based pressure sensor with a porous structure had excellent linearity over a wide detection range up to 520 kPa, which provides a good foundation for later planar pressure mapping.

Finally, the durability and stability of pressure sensors are also important in terms of practical applications. In order to study the stability of piezoresistive and capacitance response of the sensor under different strain rates and different strain degrees, nine strain cycles under different strain rates were studied and are shown in Figure 3f, and the stable sensing behavior of the sensor under different strain rates was found. The sensing performance was independent of the strain rate, which is conducive to the detection of human motion in complex motion scenes. Figure 3g shows the resistance response under a step strain. The strain ranged from 20% to 60% (increasing by 10% at each step). The initial resistance and capacitance values fluctuated slightly with each applied strain release. It was understood that when a strain removal was applied, the contact between the MWCNTs was damaged due to the recovery of the porous structure deformation, and the conductive network tended to return to its initial state. However, the applied strain could damage the microstructure of the sensor, resulting in a slight change in the resistance during compression. To demonstrate the excellent stability of the sensor, 1000 cycles of the sensor were measured at 1–520 kPa (Figure 3h) and details of the sensor response signal during the compression cycle test are shown in Appendix A. The resistance and capacitance values of the sensor remained almost unchanged after 1000 cycles, indicating that the pressure sensor had excellent durability and stability. The above data showed that the porous PDMS sensor doped with MWCNTs had a good elasticity and compressibility. ΔR/R0 and ΔC/C0 also had a good recovery rate. The results showed that the stress sensor had a high repeatability and stability.

### 3.4. Behavior Monitoring and Pressure-Sensing Arrays

As technology continues to advance and develop, more and more convenience is provided to people, but this also leaves more people in a sub-healthy state due to a lack of exercise. Therefore, in today’s wearable medical monitoring devices, it is important to monitor the human body’s behavioral signals and movement conditions at all times. Our sensors show excellent performance in this regard. The motion and human behavior of some major joints, including plantar pressure, knee motion, finger flexion, elbow motion, etc., were monitored to demonstrate the real-time motion health detection capability of these sensors as wearable electronic devices.

Due to the improvement of information technology worldwide, more and more people are using computers in their work, and the risk of cervical spondylosis caused by poor sitting posture for a long time is increasing. Posture and motion detection devices are urgently needed in people’s daily life. They can be connected to the joints of the human body and can record the movement of the flexion cycle. By detecting these signals, people’s activity status at work or in life in real time can be analyzed, and occupational diseases such as cervical spondylosis caused by sitting for too long can be prevented.

In addition, our pressure sensors could be used to track human movement. The device can be fitted to the elbow, fingers, knees and ankles and can record the body’s movements. We recorded the plantar pressure during elbow flexure (30°, 45°, 60°, 90° and 135°), finger flexure (30°, 60°, 90° and 120°) and knee joint bending (15°, 45°, 90° and 135°), as shown in Figure 4a–d. Similarly, the output signal had a good real-time response and repeatability under the different bending angles, indicating that the sensor could provide a stable real-time monitoring function in practical applications. With the increase in the bending angle of the sensor, the change in the amplitude of the output signal was significantly enhanced, and the output signal under the different bending angles could be clearly distinguished. Videos of the body-worn sensor performing the motion tests are shown in Appendix A. These results showed that the porous sensor with a 3D spherical-shell-structure MWCNT network could monitor the human motion state in real time and has great application potential in terms of medical and health monitoring and motion monitoring.

Finally, we constructed a sensor array consisting of five sensors and placed the five sensors at each of the five finger joints to detect the movement of the finger joints in order to recognize simple sign language. When different gestures were made, the sensor array received different signals due to the different changes in the finger joints. The type of sign language or gesture was detected by detecting the position and intensity of the signal changes, as shown in Figure 5b. This showed the great potential of the devices in the field of gesture recording and sign language translation. In addition, we applied the sensors to speech recognition. The strain sensor placed in the interlayer of a mask monitored the dynamics of facial muscles when people spoke and generated signal responses to the changes in the mouth shape, thus realizing speech recognition, as shown in Figure 5d.

## 4. Conclusions

In summary, we reported a strain sensor with a wide response range based on MWCNTs. To further improve the sensitivity of the sensor, we transferred a three-dimensional MWCNT structure to the internal pore surface of a porous PMDS flexible structure by wrapping MWCNTs on the surface of refined sucrose. In addition, some MWCNTs were dispersed within the PDMS structure using ultrasound. Through the above process, a porous PDMS flexible sensor with piezoresistive/capacitance dual-sensing performance was obtained, and stress changes were more accurately reflected through the changes in the capacitance and resistance. The sensor had an excellent sensing performance with a high sensitivity of 1.2/kPa and an ultra-wide response range (1–520 kPa). In addition, the sensor had a durable stability of more than 1000 loading and unloading cycles. The high sensitivity and wide detection range were explained by the sensor’s structure, which was observed by SEM and three-dimensional depth-of-field experiments. In addition, the flexibility and air permeability of the porous PDMS flexible sensor made it suitable for attaching to the surface of the human body in order to realize human posture detection, such as gesture and sign language detection. Looking ahead, these porous flexible PDMS strain sensors show potential for application in skin sensors, artificial intelligence and bionic robots.

## Figures and Tables

**Figure 1 nanomaterials-13-00843-f001:**
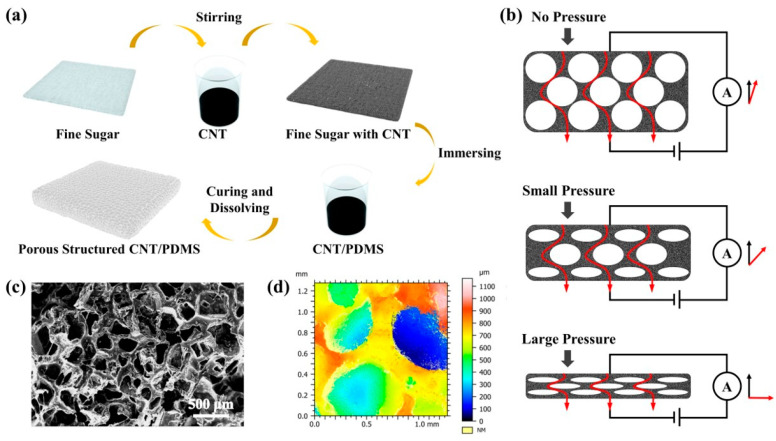
Preparation of multifunctional flexible and breathable electronic skin system with porous structure using PDMS and MWCNTs. (**a**) A diagram of the manufacturing process. (**b**) Schematic diagram of how the stress sensor works. (**c**) Porous-structure strain sensor SEM photo. (**d**) Porous-structure strain sensor 3D depth-of-field photo.

**Figure 2 nanomaterials-13-00843-f002:**
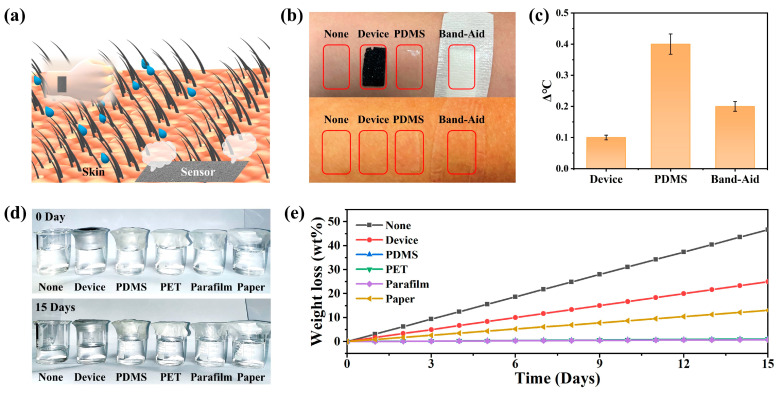
The breathability and skin-friendly nature of the pressure sensor based on the porous structure of the three-dimensional spherical-shell conductive network. (**a**) Schematic diagram showing that sensors based on porous structures could be directly connected to human skin and had skin affinity and breathability. (**b**) Changes in skin surface after different materials were adhered to arm skin for 6 h and (**c**) skin temperature changes. (**d**) Optical photos of the breathability test process at room temperature. (**e**) Water loss in glass bottles covered with films of different materials every 24 h.

**Figure 3 nanomaterials-13-00843-f003:**
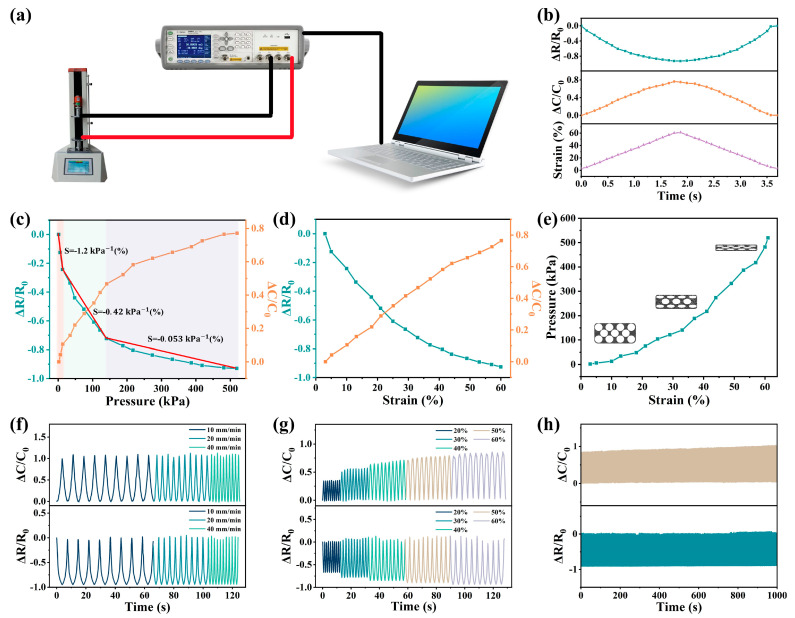
(**a**) A testing instrument used to study the piezoresistive and capacitive behavior. (**b**) The sensor had a dual-response-sensing mode that produced resistance/capacitance changes under strain. Normalized resistance and capacitance curves of the sensor in relation to (**c**) pressure and (**d**) strain. (**e**) Three strain stages of the sensor. The corresponding resistance and capacitance behaviors of sensor under (**f**) different strain frequencies, (**g**) different strains and (**h**) 1000−cycle stability test.

**Figure 4 nanomaterials-13-00843-f004:**
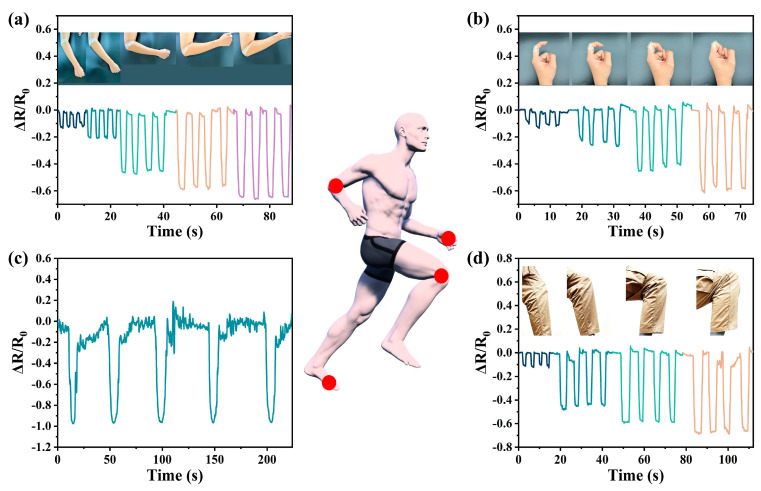
Monitoring and detection of human physiological signals and joint movements. (**a**) Detecting elbow movement. The tested motion angles of elbow joints were 30°, 60°, 90°, 120° and 150°, respectively. (**b**) Detecting the bending movement of the index finger. The tested motion angles of fingers were 30°, 60°, 90° and 120°, respectively. (**c**) Plantar pressure detection. The figure shows the significant change in plantar pressure during exercise. (**d**) Detecting knee movement. The tested knee joint motion angles were 10°, 45°, 90° and 135°, respectively.

**Figure 5 nanomaterials-13-00843-f005:**
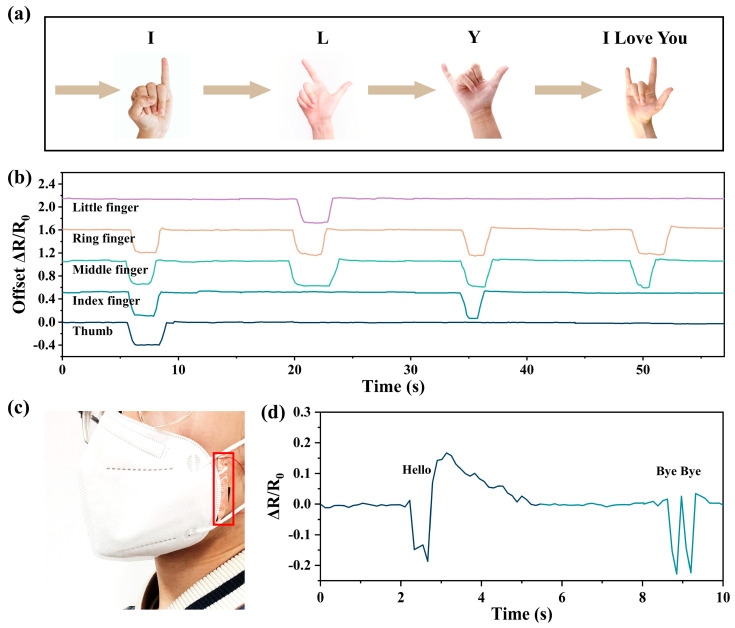
The recognition of human sign language and gestures as well as pronunciation was realized through the combination of multiple sensors. (**a**) Schematic diagram of different gestures and (**b**) resistance response signals generated by corresponding finger joint movements. (**c**) The sensor was placed on the face to recognize different articulation movements by monitoring the resistance response signal (**d**) generated by the stress change in the sensor caused by the movement of the facial muscles during articulation.

## Data Availability

No new data were created or analyzed in this study. Data sharing is not applicable to this article.

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
