# Peer review of "A Wide-Range-Response Piezoresistive–Capacitive Dual-Sensing Breathable Sensor with Spherical-Shell Network of MWCNTs for Motion Detection and Language Assistance"

_nanomaterials, 2023, doi:10.3390/nano13050843_

Round 1

Reviewer 1 Report

See attached

Author Response

We sincerely appreciate the reviewer’s comments. Please see the attachment.

Reviewer 2 Report

An applications-oriented manuscript, dedicated to the development of a simple but functional strain-sensor that minimally impact the comfort when it is directly or indirectly anchored to the skin of a human user. The sensor has a flexible porous microstructure (with open pores), and is able to function both in piezoresistive (due to the embedded multi-walled carbon nanotubes) and capacitive mode (by the means of a pair of metal electrodes which are flanking the deformable porous layer), in a wide pressure sensing range. The functional tests performed on the developed strain-sensor revealed its ability of perform well in moving, gesture and even language detection.

The major benefit to the techniques described for producing the nanoentities-based sensor consists in their ability to be scaled-up towards industrial applications, due to their simplicity.

Even it is well conceived, the manuscript has some drawbacks coming from the fact that it seems to be written in a hurry, being abundant in inaccurate and even poorly-coherent expressions. Some relevant information were also been skipped. Therefore, prior to be accepted, at least the following deficiencies should be fixed.

1. A careful revision of English language (grammar and punctuation). As examples: (i) line 57: please use "polymeric" instead of "polymer"; (ii) line 63: remove "to solve these problems"; (iii) please rephrase the paragraphs between lines 79 and 92; (iv) remove the last sentence in line 108, or rephrase it if you consider that it is carrying useful information; etc.

2.  Please specify the meaning of MWCNT acronym, once only, after the first mentioning of it.

3. In order to avoid misinterpretation, please define "breathability" or the "breathable properties" (after line 59, perhaps), in the context of comfort and related to the porous sensor you developed.

4. In subsection 2.3, please mention (and also include in subsection 2.1) the type of the elastomer and the crosslinker you have used (lines 113, 114).

5. Please insert all the details related to the equipment you mentioned in subsection 2.4 (and in all other places where relevant).

6. In the context of your discussion, in line 353, please consider to use the syntagm "piezoresistive/capacitance", instead of "piezoresistive-capacitance" hyphenated words.

Author Response

(The authors gave the same response as above.)

Reviewer 3 Report

The author proposed a strain sensor with a porous structure. However, it seems to focus on pressure type (Figure 3). The paper does not contain new findings, either theoretical or experimental. In the following section, I give specific suggestions referring to the text to improve the work's completeness, to better emphasize explicit points, and to simplify the comprehension of some concepts.

1.       The author should point out the new finding in their research. The research background was not introduced clearly. Previous literature in this research field was not sufficiently included and compared. Additional references and literature background should be added in the introduction in order to give a complete overview of the topic. Some additional information and descriptions must be added in the methodology section.

2.       Please include a comparison of the survey to those previously reported in the table.

3.       Pictures of subjects wearing the systems and performing the task could be added.

4.       The author should give more tests that produce capacitance changes under strain and pressure.

5.       In Figure 3 a, it seems the oscilloscope measure device, not LCR, please check and modify it. The schematic measurement missed the pressure and strain machine. Please check and change it. It seems to be pressure type, not strain type; please describe more detail.

Author Response

(The authors gave the same response as above.)
